# Clinical presentation, etiology, and treatment outcomes of mycetoma: A 25-year retrospective study in Southern Thailand

Sorawit Chittrakarn, Siripen Kanchanasuwan, Nattapat Sangkakul, Nonthanat Tongsengkee ®*

Division of Infectious Disease, Department of Internal Medicine, Faculty of Medicine, Prince of Songkla University, Songkhla, Thailand

* tnonthanat@gmail.com

## Abstract

### Background

Mycetoma is a chronic subcutaneous infection caused by fungi (eumycetoma) or filamentous bacteria (actinomycetoma). Although recently recognized by the World Health Organization as a neglected tropical disease, data from Southeast Asia are scarce. Previous reports from Thailand were limited and outdated.

### Methodology/Principal findings

We conducted a 25-year retrospective study (2000–2025) at a tertiary referral hospital in southern Thailand. Patients were identified from hospital records and confirmed by histopathology and/or culture. Fifty patients met inclusion criteria: 31 (62%) had eumycetoma and 19 (38%) had actinomycetoma. The median age was 50 years (IQR 41.8-58.0), and 62% were male. The foot was the most common site (80%), with bone involvement in 28%. Sinus tracts occurred in 43%, but visible grains were recorded in only 12%. Histopathology (performed in 86%) reliably distinguished fungal from bacterial etiologies, whereas culture yield was low, especially for actinomycetoma (27%). Among eumycetoma, identified pathogens included dematiaceous fungi such as *Exophiala jeanselmei* and hyaline molds such as *Scedosporium* and *Fusarium*. Among the few culture-positive actinomycetoma cases, all isolates were *Nocardia* spp. Itraconazole was the main antifungal, whereas trimethoprim-sulfamethoxazole was used for actinomycetoma. Surgery was performed in 66% of patients. At a median follow-up of 21 months (IQR 7.5–46.0), 54% achieved cure, 24% improved, 10% recurrence, and 3% required amputation.

### Conclusions/Significance

Mycetoma in southern Thailand is uncommon but clinically significant. Unlike classical endemic regions, eumycetoma predominated and was caused by diverse fungi

**Data availability statement:** All relevant data are within the manuscript and its Supporting Information files.

**Funding:** The author(s) received no specific funding for this work.

**Competing interests:** The authors have declared that no competing interests exist.

rather than *Madurella mycetomatis*. Despite combined medical and surgical therapy, cure rates were modest and complications frequent. These findings highlight regional differences in epidemiology and underscore the need for strengthened diagnostics, access to effective therapy, and region-specific neglected tropical disease strategies in Southeast Asia.

## Author summary

Mycetoma is a neglected tropical disease that causes chronic swelling, sinus tract formation, and disability. It is well known in the "mycetoma belt" of Africa, India, and Latin America, but little is known about its patterns in Southeast Asia. We reviewed 25 years of cases at a large referral hospital in southern Thailand. Fifty patients were identified, most of them middle-aged farmers. The disease usually affected the foot and sometimes spread to the bone. Unlike classical endemic regions where *Madurella mycetomatis* predominates, we found eumycetoma caused mainly by a variety of fungi, including dematiaceous molds such as *Exophiala jeanselmei*, as well as hyaline molds like *Scedosporium* and *Fusarium*. For actinomycetoma, cultures were often negative, but when positive they consistently grew *Nocardia* species. Treatment required long courses of antifungal or antibacterial drugs, often combined with surgery. Despite intensive management, many patients experienced relapse or long-term complications, and some required amputation. Our findings demonstrate that mycetoma in Thailand has distinctive causes and clinical features, differing from global trends. Recognizing these differences is important for improving diagnosis, treatment, and prevention strategies tailored to Southeast Asia.

## Introduction

Mycetoma is a chronic, granulomatous inflammatory disease of the subcutaneous tissues, progressively involving the skin, deep structures, and bone. It is broadly classified into two forms based on the causative agent: eumycetoma caused by fungi, and actinomycetoma caused by bacteria. Infection is thought to occur through traumatic inoculation of the causative organisms, commonly found in soil and on thorns, into the skin. The disease typically presents as a triad of a painless subcutaneous mass, multiple sinuses, and a discharge containing grains [1].

 Mycetoma is considered a neglected tropical disease (NTD) and continues to cause substantial morbidity and economic hardship in underserved communities. Although it is endemic in many tropical and subtropical areas called "mycetoma belt", data on its epidemiology, etiology, and management in Southeast Asia remain limited [2]. But it is reported extensively from Sudan, Mexico, and India [3]. To our knowledge, there are a total of 56 cases from SEA with the last was reported in 2012, confirmed the presence of both eumycetoma and actinomycetoma, with a wide range of

causative organisms [4]. However, in Thailand, only 31 cases were reported dating back to 1999, found that most patients were farmers from the central or northeastern regions, with *Nocardia* spp. being the predominant pathogen. Only one case originated from southern Thailand, suggesting potentially unique regional patterns [5,6]. However, these observed patterns may also be influenced by publication bias or under-reporting, given the limited number of case reports and small series available from Southeast Asia.

Recent global data further highlight this contrast. A large review of 12,379 eumycetoma cases between 2013 and 2023 found that *Madurella mycetomatis* accounted for 86% of cases, followed by *Falciformispora senegalensis* [7]. These global trends differ from the historical Thai data, emphasizing the need for updated, region-specific information. For actinomycetoma, earlier literature consistently identifies *Nocardia asteroides* and *Nocardia brasiliensis* as the predominant causative organisms [1].

Accurate identification of the causative agent is essential, as treatment differs markedly between actinomycetoma and eumycetoma. Although management is primarily guided by distinguishing fungal from bacterial etiology, species-level variation may still influence treatment response. In the absence of clinical trials, in vitro susceptibility data, animal models, and case series show varying $MIC_{90}$ values among causative fungi, suggesting potential differences in therapeutic efficacy. Timely diagnosis and appropriate therapy are critical to minimize complications and avoid unnecessary amputations [7]. Despite being an NTD, mycetoma remains neglected in regional health programs, resulting in delayed diagnosis, chronic disability, and social stigma.

Given the lack of recent data, especially from southern Thailand, this study aims to describe the clinical features, causative organisms, and treatment outcomes of mycetoma, as well as to compare the clinical characteristics of eumycetoma and actinomycetoma.

## Methods

### Ethics statement

The study was approved by the Human Research Ethics Committee, Faculty of Medicine, Prince of Songkla University (HREC number: 68-314-14-1) with waiver of informed consent due to its retrospective design. Patient confidentiality was strictly maintained, with all data anonymized prior to analysis.

### Study design and setting

This retrospective cohort study was conducted at Songklanagarind Hospital, a 900-bed university-affiliated tertiary care center in southern Thailand. The hospital serves as the main referral center for southern Thailand. The study period spanned from January 2000 to February 2025.

### Case identification and eligibility

Patients were identified through the hospital's Digital Innovation and Data Analytics (DIDA) system using ICD-10 codes consistent with mycetoma, as well as searches of histopathology reports with a diagnosis of mycetoma. Eligible patients met one of the following criteria: (i) clinical and imaging findings compatible with mycetoma, with or without microbiological evidence; or (ii) histopathological confirmation of mycetoma, with or without microbiological confirmation. Exclusion criteria were incomplete medical records, repeat episodes in the same patient (only the first episode was included), or transfer/ loss to follow-up before outcomes could be assessed.

### Data collection

Data were retrospectively extracted from electronic medical records, including outpatient and inpatient charts, microbiology and pathology reports, operative notes, and discharge summaries. A standardized case record form was used to

ensure uniform data capture. Variables collected comprised patient demographics (age, sex, occupation, geographic origin), clinical characteristics (symptom duration, trauma history, site of lesion, bone involvement, presence of pain, sinus tracts, and grains), and comorbidities such as hypertension, diabetes, malignancy, and autoimmune diseases. Diagnostic investigations, including imaging, histopathology, aerobic bacterial, anaerobic bacterial, fungal, and mycobacterial cultures, as well as molecular assays when available, were recorded. For aerobic cultures, specimens were inoculated onto blood agar, chocolate agar, MacConkey agar, and thioglycolate broth under standard aerobic conditions. Anaerobic cultures used the same media but were incubated in an anaerobic environment using the Thermo Scientific AnaeroPack-Anaero anaerobic gas generator. Fungal cultures were performed using Sabouraud dextrose agar (SDA), MY agar (SDA supplemented with chloramphenicol), and CN agar (caffeic acid agar). Microbiological results were noted to the species level where possible. Treatment details were documented, including antifungal and antibacterial regimens, duration of therapy, and surgical interventions. Outcomes of interest included cure, improvement, recurrence, progression, amputation, and complications, along with the duration of follow-up.

## Definitions

Definitions

- Eumycetoma: fungal mycetoma confirmed by histopathology, fungal culture, or molecular testing.

- Actinomycetoma: bacterial mycetoma caused by actinomycetes confirmed by histopathology or bacterial culture.

Treatment outcomes

- Cured: complete resolution of swelling and sinuses, with no discharge or new lesions for ≥3 months.

- Improved: partial clinical resolution with stable or non-progressive disease.

- Recurrence: reappearance of lesion(s) after initial improvement or cure.

- Progression: worsening of disease despite treatment.

- Required amputation: surgical removal due to advanced disease or treatment failure.

- Death due to mycetoma: mortality directly attributable to the infection or its complications.

- Death unrelated to mycetoma: mortality clearly due to other causes.

- Lost to follow-up: patients without adequate follow-up data to assess outcome.

- Unknown outcome: insufficient information in the medical record to classify the outcome.

Prognostic categories

- Favorable outcome: cured or improved, with no recurrence at last follow-up.

- Poor outcome: recurrence, progression, need for amputation, or death related to mycetoma

## Statistical analysis

Continuous variables were summarized as medians with interquartile ranges (IQR), and categorical variables as frequencies and percentages. Comparisons between eumycetoma and actinomycetoma were performed using Fisher's exact or chi-square tests for categorical variables and Mann–Whitney U tests for continuous variables. Statistical significance was defined as $p < 0.05$. Analyses were conducted using R software, Version 2024.12.1+563 (R Foundation for Statistical Computing, Vienna, Austria).

## Results

### Study population

Between January 2000 and February 2025, 154 patients were screened for eligibility (Fig 1). Of these, 102 were excluded: 86 due to incorrect diagnosis, 9 with incomplete medical records, 1 repeat episode, and 6 who were transferred or lost to follow-up. Fifty-two patients met inclusion criteria; 2 were subsequently excluded because of lack of appropriate diagnostic confirmation and the type of mycetoma could not be classified. The final analysis included 50 patients, comprising 31 cases of eumycetoma (62%) and 19 cases of actinomycetoma (38%).

### Demographic and clinical characteristics

The median age was 50 years (IQR 41.8–58.0), and 62% were male. Farmers and agricultural workers were the most common occupational group (28%). All patients originated from southern Thailand. Comorbidities were present in 36% of patients. Hypertension was the most frequent underlying disease (20%), followed by hyperlipidemia (14%) and diabetes mellitus (8%). Autoimmune disease and active solid malignancy were each reported in 6% and 4% of patients, respectively. No patients had chronic kidney disease, chronic lung disease, chronic heart disease, HIV infection, or active hematologic malignancies.

The median duration of symptoms before diagnosis was 24 months (IQR 7–60). A history of local trauma was reported in 20%, while 32% were uncertain. The foot was the predominant site of involvement (80%), followed by the hand (8%), leg/knee (4%), and arm/forearm (4%). Single cases were identified in the head/neck (2%) and chest wall (2%). No lesions occurred in the thigh, buttock, back, perineum, or abdominal wall. Eumycetoma more frequently affected the leg and hand, whereas actinomycetoma tended to involve the foot and chest wall; however, these distributions did not differ significantly between the two groups (p > 0.05).

Bone involvement was documented in 28% of cases overall, with comparable frequencies between eumycetoma (26%) and actinomycetoma (32%). Pain was reported in 52%, sinus tracts in 43%, and visible grains in 12%, without statistically significant differences between the two forms. Among the six patients with visible grains documented, grain color was recorded in only three cases: a black grain in a patient with eumycetoma (culture positive for *Exophiala jeanselmei*), a

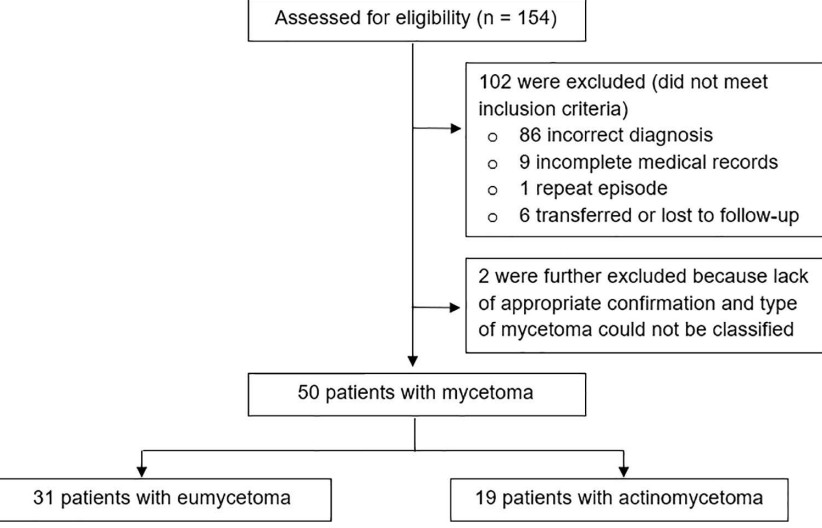

**Fig 1. Flow diagram of patient enrollment.**

yellow grain in a patient with actinomycetoma (culture positive for *Nocardia asteroides*), and a white grain in a patient with actinomycetoma, although this patient did not undergo either bacterial or fungal culture. These findings are summarized in Table 1.

### Diagnostic investigations

Among the 50 patients in this cohort, diagnostic confirmation was obtained by different modalities. Histopathology was the most frequently used, performed in 43 patients, while culture was positive in 21 patients and radiological findings supportive of mycetoma were documented in 9 patients. Diagnostic modalities often overlapped: 21 patients were diagnosed solely by histopathology, 5 by culture alone, and 1 by radiology alone. Fourteen patients had both culture and histopathology, six had histopathology with radiology, and two patients had all three modalities positive. The single patient diagnosed by radiology alone also had a positive Gram stain showing gram-positive branching bacilli, thereby allowing classification as actinomycetoma. In addition, one further patient was diagnosed exclusively by staining (positive for numerous gram-positive branching bacilli).

Histopathological examination was performed in 43 patients (86%), representing the most widely used diagnostic modality. Among eumycetoma (n = 26), fungal hyphae or elements were observed in all cases (100%), while granulomatous inflammation was present in 4 patients (15.4%) and necrosis in 1 patient (3.8%). In contrast, among actinomycetoma (n = 17), filamentous bacteria were detected in 15 patients (88.2%), and granulomatous inflammation was documented in 1 patient (5.9%). No specimens demonstrated Splendore–Hoeppli phenomenon.

Imaging studies were performed in only 9 patients (18%). Magnetic resonance imaging (MRI) was the most frequently used modality (9 cases), more common in actinomycetoma (6 vs. 3 in eumycetoma). Ultrasound was performed in one patient with actinomycetoma. No patients underwent plain radiography or computed tomography. There were no significant differences in the use of imaging modalities between eumycetoma and actinomycetoma. These results are summarized in Table 2.

Microbiological culture demonstrated variable yields (Table 3). Fungal culture was performed in 21 (70%) eumycetoma patients, of whom 18 (85.7%) yielded growth. The most frequently identified organisms were *Exophiala jeanselmei* (5 cases), *Scedosporium* spp. (3 cases, including *S. boydii*), and *Fusarium* spp. (2 cases). Other fungi included *Wangiella dermatitidis*, *Microsporum canis* (confirmed by 18S rRNA sequencing), and *Neoscytalidium dimidiatum*, as well as 5 unidentified molds. In contrast, aerobic bacterial culture was performed in 11 (58%) actinomycetoma patients, of which only 3 (27.3%) were positive, all yielding *Nocardia* spp. (*N. asteroides* in two cases). Anaerobic bacterial culture was performed in only 3 (6%) mycetoma patients, all of which yielded negative results or contamination.

Molecular diagnostic testing was performed in only one patient, whose fungal culture initially yielded an unidentified mold. Subsequent 18S rRNA sequencing identified *Microsporum canis*.

### Treatment

Among the 31 patients with eumycetoma, 30 patients (96.8%) received antifungal agents, most commonly itraconazole (28 cases, 90%). Other antifungals were prescribed less frequently, including voriconazole (3 cases), amphotericin B (2 cases), ketoconazole (1 case), and terbinafine (1 case). Combination antifungal therapy (≥2 agents) was administered in 5 patients (16.1%), with the following regimens: itraconazole plus amphotericin B (2 patients), itraconazole plus voriconazole (2 patients), and itraconazole plus terbinafine (1 patient). The median duration of antifungal therapy was 12 months (IQR 4–19). Only a minority of patients received antibiotics, typically short empirical courses. Surgical management was common, performed in 24 patients (77%). The procedures included local excision (10 patients), debridement (9), wide local excision with margin clearance (4), and incision and drainage (4). No patient required primary amputation.

In 19 patients with actinomycetoma, 17 patients (89.5%) received antibacterial agents, most commonly trimethoprim–sulfamethoxazole (TMP-SMX) (12 cases, 63%). Amoxicillin was administered in 8 patients (42%), while

**Table 1. Demographic and Clinical characteristics of mycetoma patients.**

| Variable | Eumycetoma (n = 31) | Actinomycetoma (n = 19) | Total (n = 50) | p-value |
|---|---|---|---|---|
| **Demographics** | | | | |
| Age (years), median (IQR) | 52.0 (45.5-60.5) | 47.0 (39.0-50.0) | 50.0 (41.8-58.0) | 0.109 |
| Male sex, n (%) | 22 (71.0%) | 9 (47.4%) | 31 (62.0%) | 0.171 |
| Occupation | | | | |
| • Farmer/ Agricultural worker, n (%) | 8 (25.8%) | 6 (31.6%) | 14 (28.0%) | 0.810 |
| • Office worker, n (%) | 6 (19.4%) | 3 (15.8%) | 9 (18.0%) | |
| • Housewife/ Homemaker, n (%) | 3 (9.7%) | 2 (10.5%) | 5 (10.0%) | |
| • Rubber or oil palm plantation worker, n (%) | 2 (6.5%) | 2 (10.5%) | 4 (8.0%) | |
| • Construction worker/ Manual laborer, n (%) | 2 (6.5%) | 0 (0.0%) | 2 (4.0%) | |
| • Fisherman, n (%) | 2 (6.5%) | 0 (0.0%) | 2 (4.0%) | |
| • Unknown, n (%) | 5 (16.1%) | 4 (21.1%) | 9 (18.0%) | |
| • Other [a], n (%) | 3 (9.7%) | 2 (10.5%) | 5 (10.0%) | |
| Geographic origin | | | | |
| • Southern part, n (%) | 31 (100.0%) | 19 (100.0%) | 51 (100.0%) | NA |
| Underlying disease | | | | |
| • Hypertension, n (%) | 8 (25.8%) | 2 (10.5%) | 10 (20.0%) | 0.282 |
| • Diabetes mellitus, n (%) | 4 (12.9%) | 0 (0.0%) | 4 (8.0%) | 0.284 |
| • Autoimmune disease, n (%) | 2 (6.5%) | 1 (5.3%) | 3 (6.0%) | 1.000 |
| • Active solid malignancy, n (%) | 2 (6.5%) | 0 (0.0%) | 2 (4.0%) | 0.519 |
| • Hyperlipidemia, n (%) | 4 (12.9%) | 3 (15.8%) | 7 (14.0%) | 1.000 |
| **Clinical characteristics** | | | | |
| Duration of symptoms (months), median (IQR) | 24.0 (9.5-60.0) | 18.0 (6.2-36.0) | 24.0 (7.0-60.0) | 0.580 |
| History of trauma | | | | |
| • Yes, n (%) | 7 (22.6%) | 3 (15.8%) | 10 (20.0%) | 0.234 |
| • No, n (%) | 10 (32.3%) | 11 (57.9%) | 21 (42.0%) | |
| • Uncertain, n (%) | 11 (35.5%) | 5 (26.3%) | 16 (32.0%) | |
| • Missing, n (%) | 3 (9.7%) | 0 (0.0%) | 3 (6.0%) | |
| Pain | | | | |
| • Yes, n (%) | 15 (48.4%) | 11 (57.9%) | 26 (52.0%) | 0.632 |
| • No, n (%) | 15 (48.4%) | 8 (42.1%) | 23 (46.0%) | |
| • Missing, n (%) | 1 (3.2%) | 0 (0.0%) | 1 (2.0%) | |
| Sinus tracts | | | | |
| • Yes, n (%) | 12 (38.7%) | 11 (57.9%) | 23 (46.0%) | 0.344 |
| • No, n (%) | 18 (58.1%) | 8 (42.1%) | 26 (52.0%) | |
| • Missing, n (%) | 1 (3.2%) | 0 (0.0%) | 1 (2.0%) | |
| Visible grains | | | | |
| • Yes, n (%) | 2 (6.5%) | 4 (21.1%) | 6 (12.0%) | 0.257 |
| • No, n (%) | 23 (74.2%) | 13 (68.4%) | 36 (72.0%) | |
| • Missing, n (%) | 6 (19.4%) | 2 (10.5%) | 8 (16.0%) | |
| Color of grains | | | | |
| • Black | 1 (50%) | 0 (0.0%) | 1 (16.7%) | 0.392 |
| • White | 0 (0.0%) | 1 (25.0%) | 1 (16.7%) | |
| • Yellow | 0 (0.0%) | 1 (25.0%) | 1 (16.7%) | |
| • Missing, n (%) | 1 (50%) | 2 (50.0%) | 3 (50.0%) | |

*(Continued)*

**Table 1.** (Continued)

| Variable | Eumycetoma (n = 31) | Actinomycetoma (n = 19) | Total (n = 50) | p-value |
|---|---|---|---|---|
| Site | | | | |
| • Foot, n (%) | 23 (74.2%) | 17 (89.5%) | 40 (80.0%) | 0.282 |
| • Hand, n (%) | 4 (12.9%) | 0 (0.0%) | 4 (8.0%) | 0.284 |
| • Leg/knee, n (%) | 2 (6.5%) | 0 (0.0%) | 2 (4.0%) | 0.519 |
| • Arm/forearm, n (%) | 1 (3.2%) | 1 (5.3%) | 2 (4.0%) | 1.000 |
| • Head/neck, n (%) | 1 (3.2%) | 0 (0.0%) | 1 (2.0%) | 1.000 |
| • Chest wall, n (%) | 0 (0.0%) | 1 (5.3%) | 1 (2.0%) | 0.380 |
| Bone involvement [b] | | | | |
| • Yes, n (%) | 8 (25.8%) | 6 (31.6%) | 14 (28.0%) | 0.682 |
| • No, n (%) | 22 (71.0%) | 13 (68.4%) | 35 (70.0%) | |
| • Missing, n (%) | 0 (0.0%) | 0 (0.0%) | 1 (2.0%) | |

NA; Not available

[a]Other occupations include students, retirees, and unemployed individuals

[b]Bone involvement defined by imaging confirmation or evidence of bone involvement from surgical exploration

Note: Anatomical sites with zero cases (thigh, buttock, back, perineum, and abdominal wall) were removed from the table for clarity.

**Table 2. Diagnostic investigations in patients with mycetoma.**

| | Category | Eumycetoma (n = 31) | Actinomycetoma (n = 19) | Total (n = 50) |
|---|---|---|---|---|
| Imaging, n (%) | Positive | 6 (19.4%) | 3 (15.8%) | 9 (18.0%) |
| | Not done | 25 (80.6%) | 16 (84.2%) | 41 (32.0%) |
| Histopathology, n (%) | Positive | 26 (83.9%) | 17 (89.5%) | 43 (86.0%) |
| | Not done | 5 (16.1%) | 2 (10.5%) | 7(14.0%) |
| Aerobic bacterial culture, n (%) | Positive | 0 (0.0%) | 3 (15.8%) | 4 (6.0%) |
| | Negative | 14 (45.2%) | 5 (26.3%) | 19 (38.0%) |
| | Contamination | 6 (19.4%) | 3 (15.8%) | 9 (18.0%) |
| | Not done | 11 (35.5%) | 8 (42.1%) | 19 (38.0%) |
| Anaerobic bacterial culture, n (%) | Positive | 0 (0.0%) | 0 (0.0%) | 0 (0.0%) |
| | Negative | 1 (3.2%) | 1 (5.3%) | 2 (4.0%) |
| | Contamination | 0 | 1 (5.3%) | 1 (2.0%) |
| | Not done | 30 (96.8%) | 17 (89.5%) | 47 (94.0%) |
| Fungal culture, n (%) | Positive | 18 (58.1%) | 0 (0.0%) | 18 (36.0%) |
| | Negative | 3 (9.7%) | 12 (63.2%) | 15 (30.0%) |
| | Contamination | 0 (0.0%) | 0 (0.0%) | 0 (0.0%) |
| | Not done | 10 (32.3%) | 7 (36.8%) | 17 (34.0%) |
| Mycobacteria culture, n (%) | Positive | 0 (0.0%) | 0 (0.0%) | 0 (0.0%) |
| | Negative | 17 (54.8%) | 10 (52.6%) | 27 (54.0%) |
| | Contamination | 0 (0.0%) | 1 (5.3%) | 1 (2.0%) |
| | Not done | 14 (45.2%) | 8 (42.1%) | 22 (44.0%) |
| Molecular testing [a], n (%) | Positive | 1 (3.2%) | 0 (0.0%) | 1 (2.0%) |
| | Not done | 30 (96.8%) | 19 (100%) | 49 (98.0%) |

[a]Direct PCR from clinical specimens or next generation sequencing from culture

**Table 3. Organisms identified by culture among patients with mycetoma.**

| Eumycetoma | Number (%/ fungal culture done) (n = 21) |
|---|---|
| No growth | 3 (14.3%) |
| Growth | 18 (85.7%) |
| • *Exophiala jeanselmei* | 5 (28.8%) |
| • *Scedosporium* spp. | 2 (11.1%) |
| • *Scedosporium boydii* | 1 (5.6%) |
| • *Fusarium* spp. | 2 (11.1%) |
| • *Wangiella dermatitidis* | 1 (5.6%) |
| • *Microsporum canis* [a] | 1 (5.6%) |
| • *Neoscytalidium dimidiatum* | 1(5.6%) |
| • Unidentified black mold | 3 (16.7%) |
| • Unidentified mold | 2 (11.1%) |
| Actinomycetoma | Number (%/ bacterial culture done) (n = 11) |
| No growth | 8 (72.7%) |
| Growth | 3 (27.3%) |
| • *Nocardia asteroides* | 2 (66.7%) |
| • *Nocardia* spp. | 1 (33.3%) |

[a]Unidentified mold from culture subsequently identified as *Microsporum canis* by 18S rRNA.

amoxicillin–clavulanate, rifampin, and doxycycline were each used in isolated cases. Combination antibacterial therapy (≥2 antibiotics) was given to 8 of 19 patients (42.1%), most commonly TMP–SMX plus amoxicillin (5 patients), TMP–SMX plus amoxicillin–clavulanate (2 patients), and TMP–SMX plus rifampin (1 patient). One patient received a penicillin-based multi-drug regimen. The median duration of antibiotic therapy was 14 months (IQR 7.5–19.5). Surgery was performed in 9 patients (47%), most often as local excision (5 cases) or debridement (2). These findings are summarized in Table 4.

## Outcome and complications

At a median follow-up of 21 months (IQR 7.5–46.0), 27 patients (54%) achieved cure and 12 (24%) showed improvement. Recurrence occurred in 5 patients (10%), while 2 patients (4%) experienced disease progression. One patient (2%) ultimately required amputation, and 3 patients (6%) had unknown outcomes. The median time to clinical resolution was 3.5 months (IQR 1.0–12.0), with no significant difference between eumycetoma (2.0 months, IQR 1.0–11.8) and actinomycetoma (6.0 months, IQR 1.8–12.5).

When stratified by type, outcomes were comparable: cure rates were 55% for eumycetoma and 53% for actinomycetoma. Improvement was documented in 23% and 26%, respectively, and recurrence occurred in both groups at similar rates (10%). Disease progression was observed only among actinomycetoma patients (2 cases), whereas the sole amputation occurred in a eumycetoma case.

Documented complications were uncommon. Chronic pain was reported in 2 patients (4%), functional impairment or disability in 3 patients (6%), long-term wound care in 1 patient (2%), and repeated surgeries in 3 patients (6%). Overall, 41 patients (82%) had no complications. These findings are summarized in Table 5.

## Discussion

This study represents the largest study of mycetoma reported from Southeast Asia, providing updated epidemiological, microbiological, and treatment outcome data over a 25-year period. Our findings highlight important differences in clinical presentation and causative organisms compared with both historical Thai reports and global trends.

**Table 4. Treatment of patients with mycetoma.**

| Block | Treatment | Eumycetoma (n = 31) | Actinomycetoma (n = 19) | Total (n = 50) | p-value |
|---|---|---|---|---|---|
| Antifungals | Ketoconazole, n (%) | 1 (3.2%) | 0 (0.0%) | 1 (2.0%) | 1.000 |
| | Fluconazole, n (%) | 0 (0.0%) | 1 (5.3%) | 1 (2.0%) | 0.380 |
| | Itraconazole, n (%) | 28 (90.3%) | 4 (21.1%) | 32 (64.0%) | < 0.001 |
| | Voriconazole, n (%) | 3 (9.7%) | 0 (0.0%) | 3 (6.0%) | 0.279 |
| | Terbinafine, n (%) | 1 (3.2%) | 0 (0.0%) | 1 (2.0%) | 1.000 |
| | Amphotericin B, n (%) | 2 (6.5%) | 0 (0.0%) | 2 (4.0%) | 0.519 |
| | No, n (%) | 1 (3.2%) | 14 (73.7%) | 15 (30.0%) | < 0.001 |
| Antibiotics | Penicillin, n (%) | 1 (3.2%) | 2 (10.5%) | 3 (6.0%) | 0.549 |
| | Amoxicillin, n (%) | 1 (3.2%) | 8 (42.1%) | 9 (18.0%) | 0.001 |
| | Amoxicillin-clavulanate, n (%) | 0 (0.0%) | 1 (5.3%) | 1 (2.0%) | 0.380 |
| | TMP-SMX, n (%) | 3 (9.7%) | 12 (63.2%) | 15 (30.0%) | < 0.001 |
| | Rifampin, n (%) | 0 (0.0%) | 1 (5.3%) | 1 (2.0%) | 0.380 |
| | Doxycycline, n (%) | 0 (0.0%) | 1 (5.3%) | 1 (2.0%) | 0.380 |
| | Other[a], n (%) | 0 (0.0%) | 2 (10.5%) | 2 (4.0%) | 0.140 |
| | No, n (%) | 27 (87.1%) | 2 (10.5%) | 29 (58.0%) | < 0.001 |
| Treatment duration (months, IQR) | | 12.0 (4.0-19.0) | 14.0 (7.5-19.5) | 0.560 | 12.0 (4.0-20.0) |
| Surgery | Any surgery, n (%) | 24 (77.4%) | 9 (47.4%) | 33 (66.0%) | 0.028 |
| | Incision and drainage, n (%) | 4 (12.9%) | 1 (5.3%) | 5 (10.0%) | 0.637 |
| | Debridement, n (%) | 9 (29.0%) | 2 (10.5%) | 11 (22.0%) | 0.170 |
| | Local excision, n (%) | 10 (32.3%) | 5 (26.3%) | 15 (30.0%) | 0.757 |
| | Wide local excision with margin clearance, n (%) | 4 (12.9%) | 1 (5.3%) | 5 (10.0%) | 0.637 |
| | Amputation, n (%) | 0 (0.0%) | 0 (0.0%) | 0 (0.0%) | NA |

NA; Not available

[a]Other antibiotics included clarithromycin (n = 1) and clindamycin (n = 1), both prescribed for actinomycetoma.

Note: Because none of the patients in this study received posaconazole or aminoglycoside, these rows were removed from the table for clarity.

Our cohort showed a median age of 50 years, which is notably older than the typical peak incidence of 20–40 years reported from hyperendemic areas in Sudan, Mexico, and India [8–11]. In addition, 62% of our patients were male, yielding a sex ratio lower than that described in endemic regions [3]. Prior Thai reports also indicated that males and females were equally affected [5,6]. The 2023 Thailand Agricultural Census reported that the largest farming age group was 55–64 years (accounting for 30% of farmers), and 46% were male, which may partially explain the older age distribution and the less pronounced male predominance observed in our cohort [12]. Our patients predominantly presented with foot involvement (80%), followed by hand (8%), consistent with global surveys where 80% of lesions affect the lower extremities followed by hand [13]. The duration of disease at presentation and the frequency of sinus tract formation in our cohort were comparable to previous studies; however, visible grains were documented in only 12%. This proportion was slightly lower than that reported in earlier Thai data (23%) and much lower than the 40–60% commonly reported in hyperendemic settings [5,9,14]. Several explanations are possible. First, the grain size of *Madurella mycetomatis* is usually larger and more readily observed; this species predominated in previous cohorts but was absent in our study [15]. Finally, incomplete documentation in retrospective records may have contributed to underestimation of grain visibility. Together, these factors suggest that the low frequency of observed grains in Thailand is likely pathogen-specific and partly methodological, rather than reflecting a true absence.

**Table 5. Outcomes & complications.**

| Category | Eumycetoma (n = 31) | Actinomycetoma (n = 19) | Total (n = 50) | p-value |
|---|---|---|---|---|
| Outcome | | | | |
| Favorable outcome, n (%) | 24 (77.4) | 15 (78.9) | 39 (78%) | 0.446 |
| • Cured, n (%) | 17 (54.8%) | 10 (52.6%) | 27 (54.0%) | |
| • Improved, n (%) | 7 (22.6%) | 5 (26.3%) | 12 (24.0%) | |
| Poor outcome, n (%) | 4 (12.9%) | 4 (21.1) | 8 (16%) | |
| • Progression, n (%) | 0 (0.0%) | 2 (10.5%) | 2 (4.0%) | |
| • Recurrence, n (%) | 3 (9.7%) | 2 (10.5%) | 5 (10.0%) | |
| • Required amputation, n (%) | 1 (3.2%) | 0 (0.0%) | 1 (2.0%) | |
| Unknown outcome, n (%) | 3 (9.7%) | 0 (0.0%) | 3 (6.0%) | |
| Follow-up duration (months, IQR) | 18.0 (4.0-37.0) | 28.0 (11.5-54.0) | 21.0 (7.5-46.0) | 0.277 |
| Complications | | | | |
| • Chronic pain, n (%) | 2 (6.5%) | 0 (0.0%) | 2 (4.0%) | 0.519 |
| • Functional impairment/ disability, n (%) | 2 (6.5%) | 1 (5.3%) | 3 (6.0%) | 1.000 |
| • Need for long-term wound care, n (%) | 0 (0.0%) | 1 (5.3%) | 1 (2.0%) | 0.380 |
| • Need for repeated surgeries, n (%) | 2 (6.5%) | 1 (5.3%) | 3 (6.0%) | 1.000 |
| • No complications, n (%) | 25 (80.6%) | 16 (84.2%) | 41 (82.0%) | 1.000 |

The absence of significant clinical differences between actinomycetoma and eumycetoma echoes the diagnostic challenge, and it is important to distinguish them in order to ensure that correct treatments are given [16]. In our cohort, histopathology was performed in 86% of patients and reliably differentiated between fungal and bacterial etiologies; however, it could not identify the pathogen to the species level. Although species-level variation may exist, the primary determinant of management in most clinical settings remains distinguishing fungal from bacterial etiologies. In practice, eumycetoma is generally treated with itraconazole regardless of the specific fungus, making species identification supportive but not essential for initiating therapy. Nonetheless, the true clinical impact of species-level variation remains uncertain, and further clinical trials are needed to determine whether these in vitro differences translate into meaningful differences in treatment outcomes [15]. Bacterial and fungal cultures were performed in approximately 60% of patients, but the overall recovery rate was low, particularly for actinomycetoma. Several factors likely contributed. First, many patients had received empirical antibiotics before culture, which may have suppressed pathogen growth. Second, optimal laboratory practices for actinomycete isolation such as the use of selective media, prolonged incubation, and careful handling of grains were not consistently applied, a limitation also emphasized in global reviews [16,17]. In addition, the underutilization of anaerobic culture in our center may have led to under-detection of anaerobic actinomycetes. Molecular analysis was not available locally and required expensive send-out testing, which was performed in only one case. These findings reflect a broader structural gap in Thailand and Southeast Asia, where optimized culture methods and affordable molecular diagnostics for mycetoma remain limited. Strengthening regional laboratory capacity is essential to improve pathogen identification and guide effective treatment.

Globally, *Madurella mycetomatis* is the predominant cause of eumycetoma [13]; however, it was notably absent in our series. Instead, we identified a diverse range of fungi, with *Exophiala jeanselmei* as the most common, followed by *Scedosporium* spp. and *Fusarium* spp. This pattern parallels observations from sporotrichoid lymphocutaneous infections in Southern Thailand, where dematiaceous molds predominate (*Exophiala* spp. followed by *Wangiella* spp.) [18]. In contrast, a previous review of mycetoma in Southeast Asia documented 14 cases of eumycetoma, most of which were attributed to classical agents such as *Madurella mycetomatis*, *M. tropicana, M. grisea*, and *M. mycetes*, with only a few caused by rarer

fungi (e.g., *Exophiala jeanselmei*, *Pyrenochaeta romeroi*) [4]. Our series therefore highlights a predominance of dematiaceous and opportunistic molds rather than *Madurella* species, suggesting a distinct regional epidemiology and possible ecological or exposure-related differences.

Despite combined medical and surgical therapy, the cure rate of eumycetoma in our cohort was 54%, with 10% recurrence and 3% requiring amputation. These outcomes are modestly better than those reported in Sudan, where cure rates for eumycetoma often fall below 30%, with a comparable amputation rate [19]. However, follow-up in Sudan has been particularly challenging, with up to 54% of patients dropping out during treatment. Many patients discontinue therapy and are lost to follow-up in outpatient clinics, which may partly explain why the apparent cure rate in our cohort was higher. In contrast, outcomes for actinomycetoma in our cohort were slightly lower than those reported in Sudan, where a cure rate of 63% was achieved with the use of combination TMP-SMX and aminoglycoside therapy [20]. In Mexico, a study demonstrated that the combination of TMP-SMX and aminoglycoside achieved cure rates exceeding 90% in patients with actinomycetoma [21]. In our cohort, however, no patients received aminoglycosides as part of their treatment, even among those with proven *Nocardia* spp. infections. These observations suggest that outcomes in our setting might be improved by adopting combination therapy regimens that have proven effective elsewhere.

Mycetoma is now formally recognized as a neglected tropical disease by WHO. The unusual absence of *M. mycetomatis* and predominance of dematiaceous fungi in our region parallels the pathogen patterns of other deep cutaneous infections in Thailand, suggesting unique ecological or exposure risks. These findings emphasize that mycetoma in Southeast Asia may not conform to the epidemiology observed in high-incidence African regions. Public health strategies should therefore be region-specific, emphasizing early recognition, diagnostic strengthening, and access to prolonged but effective therapy.

Strengths of this study include the largest dataset of mycetoma from Southeast Asia to date, systematic clinical and histological review, and the provision of outcome data. Nevertheless, several limitations should be acknowledged. First, the retrospective design introduced inherent risks of recall bias and incomplete documentation, particularly regarding symptom duration, trauma history, and treatment adherence. Second, reliance on medical record review meant that some demographic, clinical, and follow-up data were missing. Third, microbiological confirmation was incomplete, with culture performed in only 60% of cases and low recovery rates, especially for actinomycetoma. Fourth, molecular diagnostics and antifungal susceptibility testing were not available locally, limiting species-level identification and therapeutic guidance. Fifth, treatment regimens were heterogeneous, reflecting physician preference, drug availability, and patient financial constraints, which complicates interpretation of outcomes. Sixth, the follow-up period varied widely, and some patients were lost to follow-up, potentially underestimating relapse or long-term complications. Finally, as a single-center referral hospital study, our findings may not represent the true community burden of mycetoma in Thailand, where patients with milder disease may not be referred. As highlighted in global reviews, many of these limitations, including diagnostic gaps, incomplete microbiological confirmation, and variable treatment regimens are widespread in mycetoma research and underscore the urgent need for standardized approaches and prospective multicenter studies.

Mycetoma in southern Thailand is uncommon but clinically significant. Eumycetoma predominated, caused by diverse fungi rather than *Madurella mycetomatis*. Despite combined medical and surgical therapy, cure rates were modest and complications frequent. These findings highlight regional differences in epidemiology and underscore the need to strengthen diagnostic capacity, expand access to effective therapies, and develop regionally adapted management strategies within national and Southeast Asian neglected tropical disease programs.

## Supporting information

**S1 Dataset.** Raw dataset of 50 patients with mycetoma, including demographic variables, clinical features, microbiological results, treatments, and outcomes.
(XLSX)

## Author contributions

**Conceptualization:** Sorawit Chittrakarn, Siripen Kanchanasuwan.

**Data curation:** Sorawit Chittrakarn, Nattapat Sangkakul, Nonthanat Tongsengkee.

**Formal analysis:** Sorawit Chittrakarn, Nonthanat Tongsengkee.

**Investigation:** Sorawit Chittrakarn, Nattapat Sangkakul, Nonthanat Tongsengkee.

**Methodology:** Sorawit Chittrakarn.

**Supervision:** Sorawit Chittrakarn, Nonthanat Tongsengkee.

**Visualization:** Sorawit Chittrakarn, Nonthanat Tongsengkee.

**Writing – original draft:** Sorawit Chittrakarn, Nonthanat Tongsengkee.

**Writing – review & editing:** Sorawit Chittrakarn, Siripen Kanchanasuwan, Nattapat Sangkakul, Nonthanat Tongsengkee.

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
