## [Decision Letter · Decision Letter 0]

26 Nov 2025

Dear Dr. Tongsengkee,

Thank you for submitting your manuscript to PLOS Neglected Tropical Diseases. After careful consideration, we feel that it has merit but does not fully meet PLOS Neglected Tropical Diseases's publication criteria as it currently stands. Therefore, we invite you to submit a revised version of the manuscript that addresses the points raised during the review process.

* A rebuttal letter that responds to each point raised by the editor and reviewer(s). You should upload this letter as a separate file labeled 'Response to Reviewers '. This file does not need to include responses to any formatting updates and technical items listed in the 'Journal Requirements' section below.

* A marked-up copy of your manuscript that highlights changes made to the original version. You should upload this as a separate file labeled 'Revised Manuscript with Track Changes '.

* An unmarked version of your revised paper without tracked changes. You should upload this as a separate file labeled 'Manuscript '.

We look forward to receiving your revised manuscript.

Kind regards,

Felix Bongomin, MB ChB, MSc, MMed, FECMM

Academic Editor

Joshua Nosanchuk

Section Editor

Shaden Kamhawi

co-Editor-in-Chief

Paul Brindley

co-Editor-in-Chief

**Journal Requirements:**

At this stage, the following Authors/Authors require contributions: Sorawit Chittrakarn, Siripen Kanchanasuwan, Nattapat Sangkakul, and Nonthanat Tongsengkee. Please ensure that the full contributions of each author are acknowledged in the "Add/Edit/Remove Authors" section of our submission form.

2) We have noticed that you have uploaded Supporting Information files, but you have not included a list of legends. Please add a full list of legends for your Supporting Information files after the references list.

**Reviewers' comments:**

Reviewer's Responses to Questions

**Key Review Criteria Required for Acceptance?**

**Methods**

-Are the objectives of the study clearly articulated with a clear testable hypothesis stated?

-Is the study design appropriate to address the stated objectives?

-Is the population clearly described and appropriate for the hypothesis being tested?

-Is the sample size sufficient to ensure adequate power to address the hypothesis being tested?

-Were correct statistical analysis used to support conclusions?

-Are there concerns about ethical or regulatory requirements being met?

Reviewer #1: The authors report a case series of mycetoma in Thailand. The article is well-written and interesting. The results are relevant for these NTD. The statistical analysis is appropriate for the article.

No concerns noted regarding ethical or regulatory requirements.

Reviewer #2: The methodology is well described, although it acknowledges the limitations of a retrospective study.

Reviewer #3: As the authors mention, this is a pretty small sample size. The authors may choose to keep this article strictly descriptive and remove the statistical comparisons since the confidence intervals were so wide anyways

**Results**

-Does the analysis presented match the analysis plan?

-Are the results clearly and completely presented?

-Are the figures (Tables, Images) of sufficient quality for clarity?

Reviewer #1: Due to the sample size, the analysis and information on table 6 may be omitted.

Figure 2 could be deleted.

Reviewer #2: The presentation of the results could be improved. The tables could be better presented with the column containing the total number of cases in the rightmost column.

In Table 1, the rows containing the locations “Thigh”, “Buttock”, “Back”, “Perineum”, and “Abdominal Wall” do not need to be included in the table, since there were no cases of mycetoma of any type in these locations. Perhaps a note could be added to the legend.

Similarly, in Table 4, the rows referring to posaconazole and amikacin or gentamicin also do not need to be included in the table; it would suffice to note in the legend that no case used these medications.

Figure 2, which shows the overlap of diagnostic approaches, was not immediately clear: 21 patients underwent culture, but in the figure, the number 21 also identifies the 21 patients who underwent only histopathology, which can be confusing. Perhaps a stronger colored line could be added around the circles corresponding to the tests with numbers in the legend.

Reviewer #3: (No Response)

**Conclusions**

-Are the conclusions supported by the data presented?

-Are the limitations of analysis clearly described?

-Do the authors discuss how these data can be helpful to advance our understanding of the topic under study?

-Is public health relevance addressed?

Reviewer #1: The conclusions and limitations are adequate. The authors also discuss potential ways to improve care.

Reviewer #2: The conclusions are based on the data presented and the limitations of the research are described. The authors highlight that the etiological profile of the disease identified in the country, different from other areas, may demand strategic control actions and differentiated therapies with great relevance to public health.

Reviewer #3: Lines 337-338: Is it necessary to identify to species level to guide therapy? In most settings, eumycetoma (for example) will be treated with itraconazole, regardless of causative fungus. While species level identication can be helpful, it may not be critical in managing a patient. Rather distinguishing between bacteria and fungi is vital.

Lines 360-361: Phialophora jeanselmei is just an older, updated name for Exophiala jeanselmei

Lines 383-385: The mycetoma belt incldues parts of central and south America in which other organisms other than M. mycetomatis almost predominate. Perhaps the authors should point out that the epi differs from high incidence areas on the African continent instead.

**Editorial and Data Presentation Modifications?**

Reviewer #1: Minor review.

Reviewer #2: (No Response)

Reviewer #3: (No Response)

**Summary and General Comments**

Reviewer #1: The manuscript is well-written.

Please include in the abstract the IQR ranges.

Please detail the culture media used in the center.

Please detail if any patients had a combination antibiotic or antifungal treatment.

Table 6 could be deleted.

Figure 2 could be deleted.

Reviewer #2: This is a relevant study that fills a knowledge gap regarding mycetoma in Southeast Asia, focusing on Thailand. Due to the rarity of the disease and its status as a Neglected Tropical Disease, prospective controlled studies are difficult to conduct. Within the methodological possibilities, the authors managed to extract important information, identifying the peculiar profile of the etiological agents in the country, which may contribute to new therapeutic approaches. The authors provided a good analysis of the research limitations. For these reasons, I would strongly recommend publication with minor adjustments to the presentation of the results.

Reviewer #3: Thank you for the opportunity to review this retrospective study on mycetoma in Southern Thailand. These data are critical to better understand the epidemiology this NTD in Southeast Asia. Some comments and thoughts below:

- Lines 78-79: While it could be unique regional patterns, it might also be due to gaps in reporting. Consider adding a note about publication bias.

- Lines 80-83: The authors mention eumycetoma causative agents from the large literature review, but what about actinomycetoma? Consider listing common agents from that review as well

- Lines 85: Is identification of causative agent essential? Or just etiology? Eumycetoma is likely to be treated the same regardless of causative fungi and same with actinomycetoma. The most important aspect is differentiating between fungi vs bacteria. Consider clarifying

PLOS authors have the option to publish the peer review history of their article (what does this mean? ). If published, this will include your full peer review and any attached files.

**Do you want your identity to be public for this peer review?** For information about this choice, including consent withdrawal, please see our Privacy Policy .

Reviewer #1: No

Reviewer #2: No

Reviewer #3: **Yes:** Dallas Smith

**Figure resubmission:**
---

## [Editor Report · Decision Letter 1]

3 Dec 2025

Dear Mr Tongsengkee,

We are pleased to inform you that your manuscript 'Clinical presentation, etiology, and treatment outcomes of mycetoma: A 25-year retrospective study in Southern Thailand' has been provisionally accepted for publication in PLOS Neglected Tropical Diseases.

Best regards,

Joshua Nosanchuk, MD

Section Editor

Joshua Nosanchuk

Section Editor

Shaden Kamhawi

co-Editor-in-Chief

Paul Brindley

co-Editor-in-Chief

---

## [Editor Report · Acceptance letter]

Dear Mr Tongsengkee,

We are delighted to inform you that your manuscript, "Clinical presentation, etiology, and treatment outcomes of mycetoma: A 25-year retrospective study in Southern Thailand," has been formally accepted for publication in PLOS Neglected Tropical Diseases.

Best regards,

Shaden Kamhawi

co-Editor-in-Chief

Paul Brindley

co-Editor-in-Chief
